# Spatiotemporal co-distribution and time lagged cross correlation of malaria and dengue in Loreto, Peru

Gabriel Carrasco-Escobar[1,2], Paloma M. Carcamo[1,3]*, Samantha R. Kaplan[4], Jesus M. Quispe[1], Gordon C. McCord[5], Tarik Benmarhnia[2]

1 Health Innovation Laboratory, Alexander von Humboldt Institute of Tropical Medicine, Cayetano Heredia University, Lima, Peru, 2 Scripps Institution of Oceanography, University of California San Diego, La Jolla, California, United States of America, 3 Yale School of Public Health, Yale University, New Haven, Connecticut, United States of America, 4 Division of Infectious Diseases, School of Medicine, University of Colorado, Aurora, Colorado, United States of America, 5 School of Global Policy and Strategy, University of California San Diego, La Jolla, California, United States of America

☉ These authors contributed equally to this work.
* paloma.carcamo.g@upch.pe

## Abstract

Malaria and dengue account for most vector-borne disease-related cases and deaths worldwide, disproportionately affecting tropical regions such as Peru. Previously identified social, environmental, and climate determinants for both diseases are similar despite differences in vector ecologies. Control strategies for both rely on interventions such as removal of breeding sites or insecticide-based strategies, which could be integrated. We assessed synchrony (temporal correlations, temporal order, lagged relationships) and spatial correlations between malaria and dengue incidence in the Loreto region of Peru. We conducted a time-lagged cross-correlation (TLCC) analysis between district-level dengue and malaria time series in Loreto between 2000–2021. We identified temporal patterns of dengue that could precede malaria patterns or vice versa. We grouped districts based on shared temporal and spatial patterns in dengue and malaria incidence and explored how the two diseases co-occurred geographically. Our analysis shows a growing number of districts reporting both dengue and malaria over time. Maximum TLCC coefficients varied in magnitude and direction between districts, as did corresponding lag times. In the Northwest, increases in malaria often preceded increases in dengue, while in the Northeast increases in malaria preceded decreases in dengue cases. We found spatial correlation between coefficients in some regions in the Northwest, suggesting that characteristics of a geographic area may influence the observed associations. The identification of districts with strong associations between dengue and malaria incidence can inform implementation of targeted integrated interventions, while identification of distinct patterns of association can inform future studies assessing drivers of both diseases in different settings.

**Data availability statement:** All the data and code for full reproducibility of the analyses in this paper are available in the following repository: https://github.com/healthinnovation/loreto_synchrony.

**Funding:** This work was supported by Wellcome Trust (226428/Z/22/Z to GCE). The funders had no role in study design, data collection and analysis, decision to publish, or preparation of the manuscript.

**Competing interests:** The authors have declared that no competing interests exist.

## Introduction

Mosquito-borne diseases are a critical cause of morbidity and mortality worldwide. The World Health Organization estimates that about 80% of the world's population is at risk of vector-borne diseases (VBD), with a disproportionate burden falling on tropical and subtropical regions [1], and increasing geographical areas becoming affected in the context of climate change and variability. Malaria and dengue account for the majority of VBD-related cases and deaths worldwide, causing an estimated yearly 212 and 96 million cases, and 429 and 9 thousand deaths respectively [1]. The Latin American region has one of the highest burdens of VBD, and socioeconomic and health access inequities make eradicating these diseases in the region uniquely challenging [2]. Such is the case of Peru, an ecologically diverse country encompassing coastal deserts, Andean highlands, and tropical rainforests, which has experienced increases in VBD incidence in recent years [3,4].

Malaria has been present in Peru since at least the 1500s and has been a significant cause of morbidity and mortality in the coast and jungle regions. Eradication interventions have not been successful [5,6]; currently, malaria is still endemic in the jungle, with 26 368 cases reported in 2022, 83·7% of them in the Loreto region [7]. Cases are reported year-round, peaking around the rainy season between November and April. Transmission has also been linked to deforestation and human incursions into forested areas for logging, hunting, or other economic activities [6]. The primary vector in Peru is *Anopheles (Nyssorhynchus) darlingi*, which is more abundant in rural settlements [8].

Dengue was first identified in Peru in 1990 during an outbreak in the Loreto region, and there have been recurrent annual outbreaks of varying magnitude since, with a general increasing trend in cases [7]. In 2023, an unprecedented 270 978 cases were reported up to epidemiological week 45, a 486% increase compared to the same period in 2022 [9]. Control measures are generally aimed at removing potential breeding sites for the vector; insecticide spraying is reserved for outbreaks, as there is considerable resistance [10]. As with malaria, dengue cases are confined to the lowlands of the country where climate and land cover are suitable for the vector, *Aedes aegypti*, a predominantly urban-dwelling mosquito [11]. In the jungle, outbreaks generally occur during the rainy season [12,13]. Meanwhile, in the coastal region, outbreaks are sporadic and usually related to introductions from the jungle under favourable climate conditions during El Niño events [13,14]. Climate-related drivers of disease transmission have not yet been fully elucidated but previous epidemiological evidence has shown how higher temperatures can increase outbreak risk [13,14].

While these two VBD share many similarities in their social, environmental and climate determinants, as well as which communities are disproportionately impacted, the epidemiological literature has studied them separately, without assessing potential spatial and temporal relationships between them. Furthermore, despite control strategies for both diseases relying on similar interventions (e.g., early warning systems, removal of breeding sites, limiting outdoor activities, and targeted health

campaigns), these programs are not integrated in Peru or in many other endemic countries. Understanding whether the dynamics of dengue and malaria are related in time or space could help to align surveillance and response activities, improving the efficiency of public health programs in resource-limited settings.

In this study, we propose a framework to test whether incidence peaks of one disease precede or follow those of the other, after accounting for temporal autocorrelation and seasonality. This is important because common drivers such as rainfall, temperature anomalies, or land use change could produce synchronous or sequential epidemics. If consistent lagged relationships exist, they could be used to strengthen early warning systems, for example by anticipating dengue outbreaks from preceding malaria signals or vice versa. Our study focuses on Loreto, a department in the Peruvian Amazon that is endemic for both dengue and malaria. Our primary research question was whether the temporal dynamics of dengue and malaria are correlated at the district level, and if so, whether one disease tends to lead or lag the other. We hypothesized that, because both diseases share climatic and environmental drivers, their time series would exhibit significant correlations, but that the direction and timing of these relationships might vary by district due to differences in vector ecology, urbanization, and human mobility. We further asked whether these relationships display spatial clustering, which would suggest common underlying processes at the landscape level. By clarifying synchrony (i.e., correlations without lag), temporal ordering (which disease tends to precede the other), and lag structure, our aim was to evaluate whether surveillance and control strategies for dengue and malaria in Loreto could be more effectively integrated.

## Methods

### Study design

We conducted a retrospective analysis using secondary data of weekly dengue and malaria cases in Loreto, Peru from 2000 to 2021 to analyse synchrony and spatial correlation in their incidence.

### Study area

Loreto is located in Northeastern Peru and is the largest and second most sparsely populated department in the country, with an area of 368 799·5 km² and a population of about one million [15,16]. It is part of the Amazon rainforest, comprising areas of high and low altitude jungle. Administratively, it is divided into eight provinces and 53 districts (**Fig 1A**). More than half the population lives in the district of Maynas, where the capital city of Iquitos is located; most other districts have low population densities with small semi-urban or rural communities [17]. Climate conditions are warm and humid; temperatures fluctuate between 16–35°C and mean monthly precipitation levels are close to 250 mm, peaking at approximately 400 mm during the rainy season (November-March) [17]. Conditions in the area are suitable year-round for various mosquitoes of epidemiological importance, including those of genuses *Aedes, Anopheles* (*Nyssorhynchus*), *Haemagogus*, and *Culex* [18,19]. The region has a high burden of VBD including malaria and dengue, with sporadic cases of Zika, chikungunya, Mayaro, and yellow fever [20].

### Data sources

Individual laboratory confirmed dengue and malaria case reports, both subject to mandatory reporting in Peru, were obtained from the Loreto Regional Health Department (Gerencia Regional de Salud Loreto, Geresa Loreto) through an official data request on December 1, 2023. These surveillance data are routinely collected for public health purposes and are not directly downloadable from an open database but can be requested through Peru's public information access system (https://www.gob.pe/20399-solicitar-acceso-a-la-informacion-publica).

These datasets include date of symptom onset for all reported cases. Cases with symptom onset between January 1, 2000 and December 31, 2021 were included in the analysis. For descriptive visualization in **Fig 1**, cases for the entire Loreto region were aggregated by month of symptom onset. For the primary analysis, data were aggregated to district and

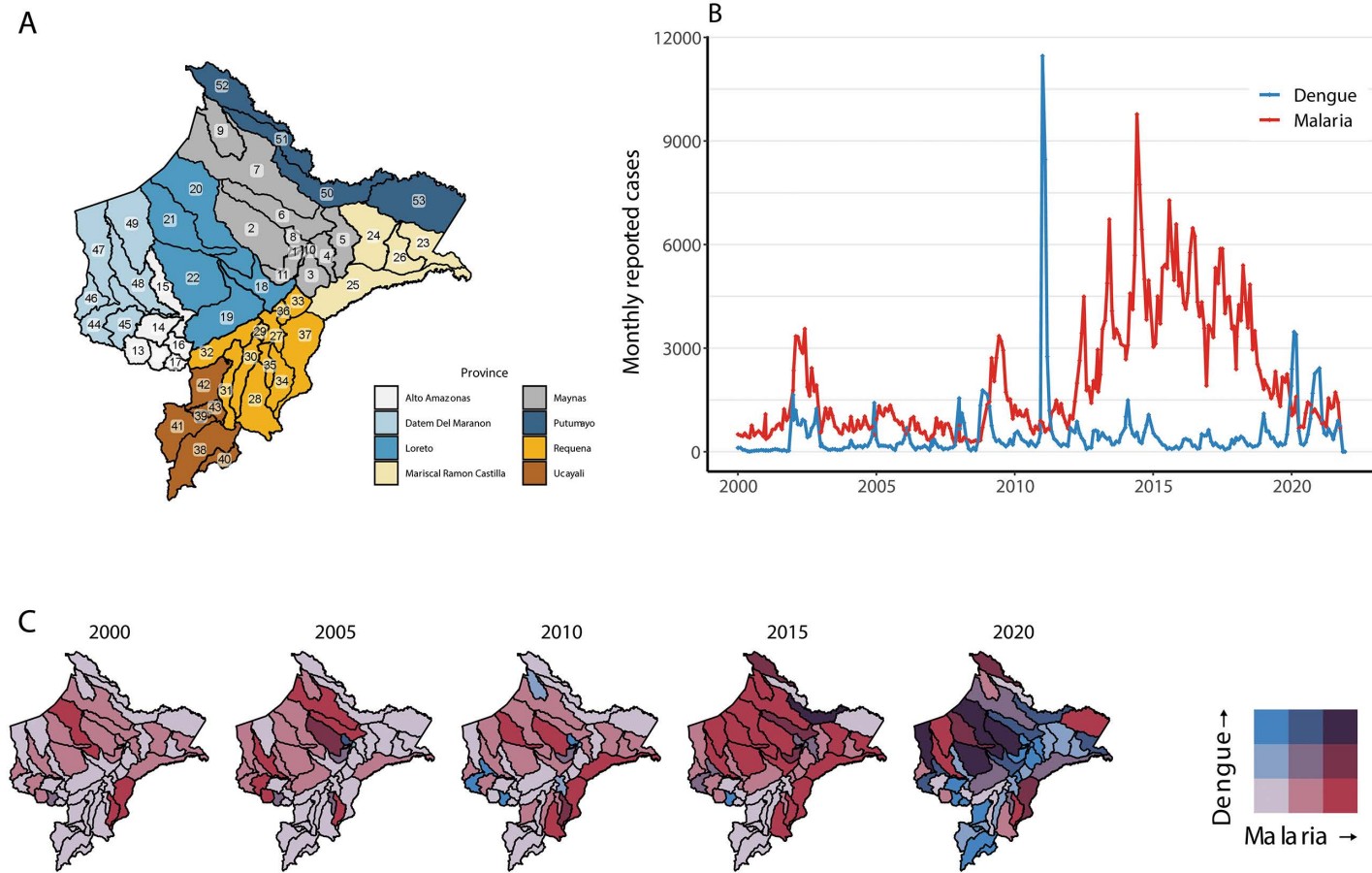

**Fig 1. Dengue and malaria spatio-temporal patterns in Loreto. (A)** Districts* and provinces of Loreto department in Peru. **(B)** Reported monthly cases of malaria and dengue in Loreto between 2000 and 2021. **(C)** Dengue and malaria incidence (log-incidence categorised using Fisher-Jenks algorithm for the entire study period) by district in Loreto (see **S2 Fig** for maps for all study years). Maps produced in R v.4.5 [21] using public data from Instituto Nacional de Estadística e Informática (INEI - Peru) contributors (https://estadist.inei.gob.pe/map) under Open Data Commons Open Database License (ODbL) 1.0 (http://openstreetmap.org/copyright). *1 Iquitos, 2 Alto Nanay, 3 Fernando Lores, 4 Indiana, 5 Las Amazonas, 6 Mazan, 7 Napo, 8 Punchana, 9 Torres Causana, 10 Belen, 11 San Juan Bautista, 12 Yurimaguas, 13 Balsapuerto, 14 Jeberos, 15 Lagunas, 16 Santa Cruz, 17 Teniente Cesar Lopez Rojas, 18 Nauta, 19 Parinari, 20 Tigre, 21 Trompeteros, 22 Urarinas, 23 Ramon Castilla, 24 Pebas, 25 Yavari, 26 San Pablo, 27 Requena, 28 Alto Tapiche, 29 Capelo, 30 Emilio San Martín, 31 Maquia, 32 Puinahua, 33 SAquena, 34 Soplin, 35 Tapiche, 36 Jenaro Herrera, 37 Yaquerana, 38 Contamana, 39 Inahuaya, 40 Padre Marquez, 41 Pampa Hermosa, 42 Sarayacu, 43 Vargas Guerra, 44 Barranca, 45 Cahuapanas, 46 Manseriche, 47 Morona, 48 Pastaza, 49 Andoas, 50 Putumayo, 51 Rosa Panduro, 52 Teniente Manuel Clavero, 53 Yaguas.

epidemiologic week based on usual residence and date of symptom onset. District boundaries from the 2017 census were used. District-weeks without reported cases were considered to have zero cases. District-level population estimates were obtained from the Peruvian National Statistics Institute. For newly created districts without estimates for the beginning of the study period, the first available population estimate was used for all previous years.

### Statistical analyses

Yearly incidence per 1000 inhabitants per district was calculated using reported cases and population estimates. To describe spatial co-occurrence of dengue and malaria, we classified log-incidence values using the Fisher-Jenks algorithm with three classes, applied to the pooled distribution over the entire study period to ensure year-to-year

comparability, as implemented in the cartography R package. These fixed boundaries were used to map annual bivariate categories.

We conducted a time-lagged cross-correlation (TLCC) analysis between dengue and malaria time series for Loreto. Because TLCC assumes stationary series, we applied first differences to both weekly case series. Stationarity of the differenced malaria and dengue series was evaluated using the Augmented Dickey-Fuller test, which indicated that all district-level series were stationary (all $p < 0.01$, **S1 Fig**). Residual autocorrelation was assessed with Ljung-Box tests (lag 52), which showed that some districts still had annual periodicity. Therefore, we applied a lag-52 difference to each weekly time series to remove annual seasonality, after which Ljung-Box tests all yielded p-values $< 0.01$. We then used a TLCC function on the seasonally adjusted time series to identify lag patterns of dengue time series that could be predecessors for malaria time series or vice versa. In our setting, positive lags meant that malaria cases preceded dengue cases, while negative lags meant that dengue cases preceded malaria cases. We assessed lags in a time window of ±2 years (104 weeks); we considered this the maximum informative period to observe associations between the two diseases. Optimal lags were defined as those corresponding to the maximum absolute cross-correlation coefficient for each district. Cross-correlations were assessed at the district level. After extracting maximum TLCC coefficients and their respective optimal lag for all districts, we conducted Moran's global and local tests for spatial autocorrelation of both metrics. We categorised districts according to their specific dengue/malaria spatiotemporal patterns. As sensitivity analyses, we first repeated our analyses using time windows of 1–5 years since dengue outbreaks in the Americas follow a cyclical pattern, recurring every 3–5 years. Additionally, we conducted bootstrapping by randomly reassigning time indexes to observations within a district, and, separately, randomly generating the case counts (preserving each district's case count mean and dispersion parameters assuming a negative binomial distribution). For each iteration (999 replicates), we repeated the TLCC analysis and then compared the distribution of TLCC coefficients and associated lags from the reshuffled dataset to those observed in the original dataset using a Kolmogorov-Smirnov (K-S) test. These data randomization tests help assuage concerns that correlations between dengue and malaria case time series may be spurious. Statistical significance threshold for all tests was set at $\alpha = 0.05$. All analyses were performed using R (R version 4·5 for Windows) [21].

### Ethics

This study used routinely collected surveillance data from the Loreto Regional Health Department. This dataset was fully anonymized prior to being shared and contained no personal identifiers or sensitive information. Because data were de-identified and collected as part of routine public health surveillance, the Universidad Peruana Cayetano Heredia Institutional Committee for Ethics in Research waived the requirement for participant consent (approval number 442-41-23, October 13, 2023).

### Results

A total of 130 597 dengue and 531 376 malaria cases were reported in Loreto between 2000–2021 and included in analyses (**Fig 1B**). Dengue cases peaked in 2011. The district with the highest number of reported cases was Iquitos, in central Loreto, with 10 456 cases in 2011. Teniente Manuel Clavero, the northernmost district, reported the highest yearly incidence of 119 cases per 1000 in 2019. Malaria cases peaked in 2017. The district with the most reported cases was Andoas in Northwestern Loreto with 10 680 cases in 2017. Soplin had the highest yearly incidence of 1117 cases per 1000 in 2009 (**S2–S4 Figs**).

Descriptive analysis of the spatio-temporal distribution of dengue and malaria cases showed a shift from primarily malaria incidence in earlier years to incidence of both diseases in most districts in later years, especially in the Northwest (**Fig 1C**, **S5 Fig**). The two diseases were jointly present in only fifteen districts in 2000: Yurimaguas, Nauta, Parinari, Tigre, Urarinas, Pebas, Ramon Castilla, Alto Nanay, Belen, Iquitos, Mazan, Punchana, San Juan Bautista, Requena, and Contamana, most of them located in central Loreto. The number of districts reporting co-occurrence markedly increased

over time due to increases in both malaria and dengue incidence; by 2021, an additional thirty-three districts reported both diseases, with only one district (Contamana) not reporting co-occurrence anymore. In the first five study years, only six districts along Northeastern Loreto (Pampa Hermosa, Padre Marquez, Vargas Guerra, Saquena, San Pablo, Requena) had at least one year with only dengue cases; by the last five years, the only one of these that did not report co-occurrence was Padre Marquez. In the case of malaria, thirty-eight districts reported only malaria in at least one year between 2000–2004, and all but two of these (Cahuapanas, Padre Marquez) reported co-occurrence in the five final years. Notably, towards the end of the study period, several districts along the Southeastern border of the region reported significant increases in dengue incidence when compared to earlier years.

Results of TLCC analysis for eight representative districts are shown in Fig 2 (S6 Fig shows results for all districts). Correlation coefficients varied both in magnitude and direction between districts, as did lag times. Twenty-seven districts had positive coefficients, indicating positive association between dengue and malaria cases. Coefficients ranged from 0·03 to 0·87 (mean 0.20, standard deviation [SD] 0.17), and lags varied from -104–103 weeks (mean 19.15, SD 62.35) (S7 Fig shows a map of maximum TLCC coefficients and corresponding lag times, as well as the distribution for both). Andoas, in Northwestern Loreto, had the highest positive correlation coefficient of 0·87 at a lag of 103 weeks, indicating that malaria cases typically preceded dengue cases by 103 weeks. Twenty-five districts had negative coefficients, indicating a negative association (i.e., increases in malaria cases were associated with decreases in dengue cases). Negative coefficients ranged from -0·04 to -0·35 (mean -0.13, SD 0.07), and lags ranged from -95–100 weeks (mean 12.92, SD 70.60). Yaguas, in the Northeastern border, had the lowest coefficient at -0·35 with a lag of 82 weeks, meaning that malaria cases were associated with a decrease in dengue cases after 82 weeks. No dengue cases were reported in Rosa Panduro during the study period, thus coefficients were not calculated. Districts in Northwestern Loreto tended to have more positive correlation coefficients. Of the ten districts with highest coefficients, five (Teniente Manuel Clavero, Tigre, Andoas, Mazan, Lagunas) were located in Northwestern Loreto, along the border with Ecuador. Among them, coefficients varied from 0·19 to 0·87, and lag times from -59–103 weeks. On the other hand, six of the ten districts with larger negative coefficients (Alto Tapiche, Tapiche, Soplin, Yaquerana, Jenaro Herrera, Yaguas) were located along the Southeastern border, with coefficients ranging from -0·35 to -0·15 and lags from -93–100 weeks (Fig 3A).

We conducted sensitivity analyses (S8–S10 Figs) to assess for consistency in observed correlation coefficients and lag times when using different time windows. Although both metrics varied when time windows were changed, coefficients were consistently positive or negative in 31 of the 52 districts (S8 Fig). Bootstrapping sensitivity analyses revealed that the original distribution of TLCC coefficients was significantly different from the observed distribution after randomly generating case counts, as well as after randomly reassigning time indices (p<0.01 for both). The distribution of corresponding lag times was significantly different with randomly generated case counts (p<0.01), but not with reassigned time indices (p=0.08).

A global Moran's test showed no overall spatial autocorrelation for the study area (Moran's *I* statistic 0.02, p=0.32). However, a local Moran's test showed some significant spatial autocorrelations (Fig 3B). Two Northwestern districts (Trompeteros, Urarinas) had positive TLCC coefficients and were primarily surrounded by other districts with positive coefficients. Pastaza, which had a negative TLCC coefficient, had a negative test statistic, indicating that it was primarily surrounded by districts with positive coefficients. Local Moran's test statistics for all other districts were not statistically significant. We also conducted a local Moran's test for associations between lag times (Fig 3C). Only Yavari, in the northeast, and Tigre, in the northwest, had significant positive spatial autocorrelation.

## Discussion

In this study, we aimed at advancing a novel framework to investigate the spatio-temporal relationships between malaria and dengue in Peru and categorise geographical regions according to these patterns. Such an analytical framework can be particularly helpful to inform integrated strategies in areas where both of these climate-sensitive diseases are increasing in incidence and may interact exponentially.

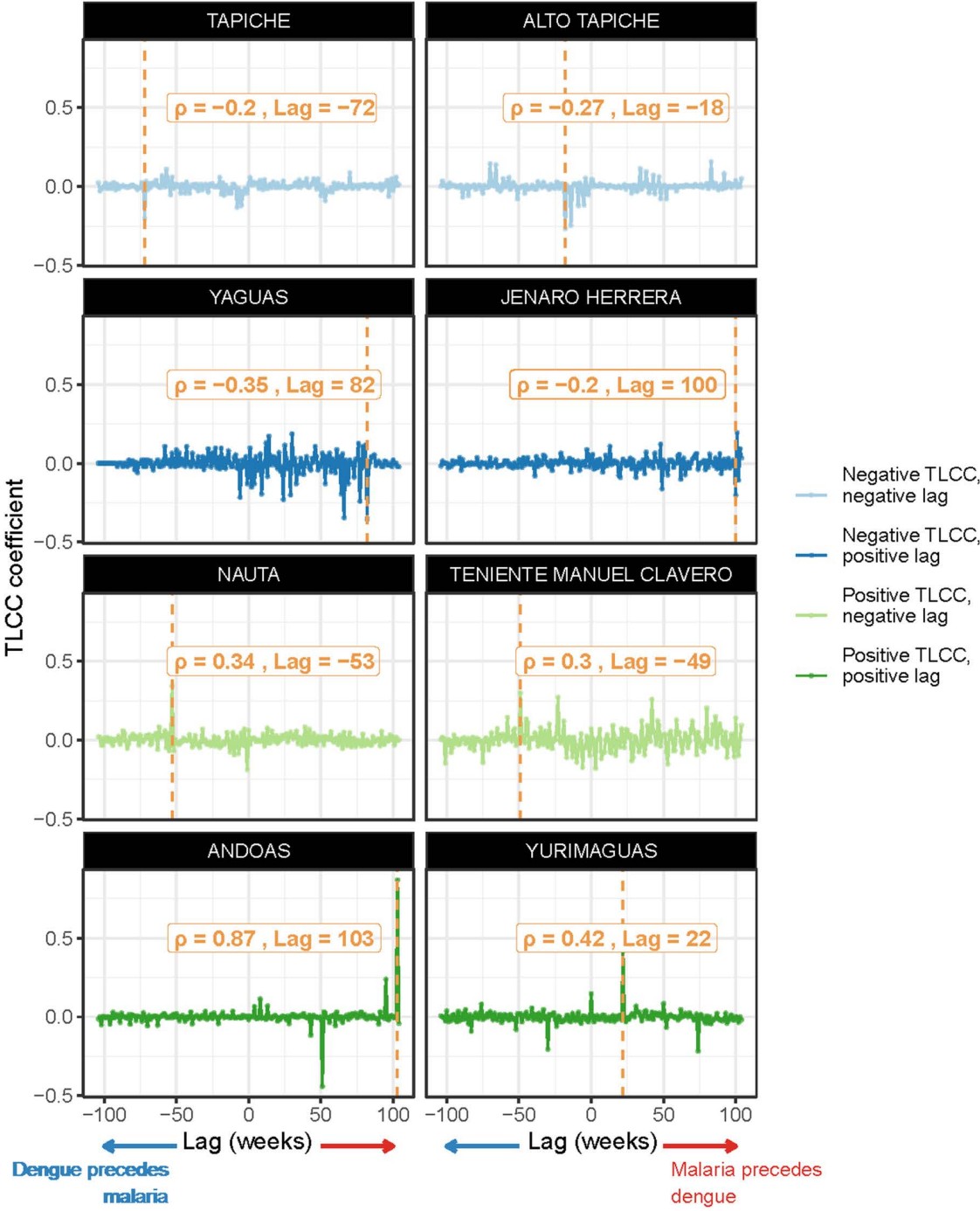

**Fig 2. Results of TLCC analysis: maximum absolute coefficients and corresponding lags in weeks.** TLCC analysis of dengue and malaria time series at lags ±2 years for selected districts in Loreto: Tapiche, Alto Tapiche, Yaguas, Jenaro Herrera, Nauta, Teniente Manuel Clavero, Andoas and Yurimaguas (districts with the highest coefficient in each category). ρ indicates the maximum correlation coefficient and lags are in weeks; the lag of the maximum correlation coefficient is marked with a vertical dashed orange line. Plots for each district are included in **S6 Fig**.

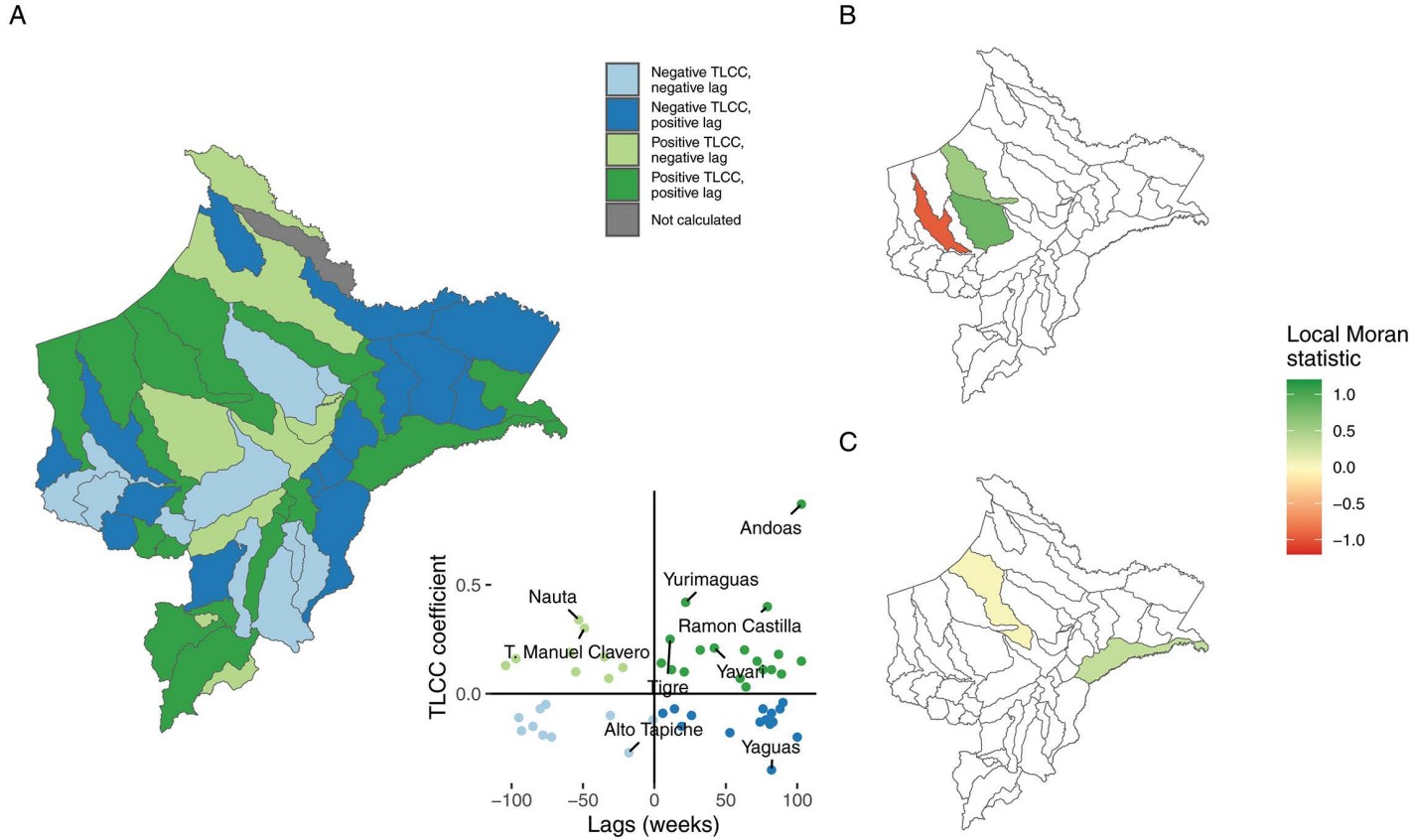

**Fig 3. Categorisation of districts based on type of correlation observed and results of Moran's local test for spatial correlation of coefficients and lags. (A)** Categories of TLCC analysis results for districts in Loreto. Inset shows scatter plot of TLCC coefficient and lags in weeks, districts with coefficients higher than 0.2 are labelled. **(B)** Results of Moran's local test for spatial correlation of coefficients. Districts in green have a positive test statistic, indicating they are primarily surrounded by districts with similar TLCC coefficients. Districts in red have a negative test statistic, indicating they are primarily surrounded by districts with different TLCC coefficients. Districts in white did not have a significant test statistic. **(C)** Results of Moran's local test for spatial correlation of lags. Districts in green have a positive test statistic, indicating they are primarily surrounded by districts with similar lag values. Districts in red have a negative test statistic, indicating they are primarily surrounded by districts with different lag values. Districts in white did not have a significant test statistic. Maps produced in R v.4.5 [21] using public data from Instituto Nacional de Estadística e Informática (INEI - Peru) contributors (https://estadist.inei.gob.pe/map) under Open Data Commons Open Database License (ODbL) 1.0 (http://openstreetmap.org/copyright).

We found an increase in the number of districts experiencing both dengue and malaria in Loreto over the 22-year study period, predominantly in the Northwestern region. Analysis of time series of both diseases showed heterogeneous temporal associations across the study area. In districts located in the Northwest, increases in malaria cases preceded increases in dengue cases, while in others located in Northeast, increases in the incidence of malaria led to a decrease in dengue. Additionally, the timing of these associations showed considerable variability. We found spatial correlation between positive coefficients in Trompeteros and Urarinas and between negative coefficients in Pastaza, suggesting that specific small-scale environmental or geographic characteristics and patterns of land use may influence the association between dengue and malaria. One plausible explanation is that stages of urbanisation and deforestation can differentially affect exposure to vectors and disease risk: early stages of forest clearing may increase exposure to both *An. darlingi* and *Ae. aegyti,* while more advanced settlements may be most suitable for *Ae. aegypti* only. In addition, the proximity of protected areas, which cover about 20% of the total area of Loreto, may affect observed spatial patterns by limiting land conversion [22,23]. Although the timing and directionality of the observed associations displayed marked variability, these

findings can inform implementation of targeted integrated interventions for malaria and dengue in districts in which strong associations were observed. Furthermore, the identification of four distinct patterns of association between dengue and malaria can be used to inform future studies assessing specific modifiers of these associations in each specific area.

In districts where increases in malaria preceded dengue, malaria surveillance could serve as an early signal for anticipating dengue outbreaks, providing an operational window to mobilize dengue-specific resources before incidence rises. Identifying these patterns can guide the timing of activities such as insecticide spraying, larval source reduction, and health communication campaigns, ensuring that resources are deployed at the moment they are most likely to be effective. Spatial clustering of similar dengue–malaria association patterns further suggests that integrated strategies could be planned at the subregional level, rather than on a district-by-district basis. For example, districts in Northwestern Loreto with consistent positive associations might benefit from coordinated, simultaneous campaigns targeting both vectors.

Changes in environmental and climate conditions, including climate change and variability, deforestation, and urban development, as well as changes in population mobility patterns, have led to shifts in transmission dynamics of VBD [24]. A clear and concerning example of this is the increase in reports of dengue-malaria co-infection both in Peru [25] and other tropical settings [26–31]. Thus, it becomes important to elucidate the patterns of co-occurrence of these two diseases in order to mount appropriate prevention and control strategies.

In considering potential causes for this co-occurrence, it is important to consider the ecology of their vectors to identify possible overlapping drivers of abundance or changes in behaviour. Despite marked differences in ecological niches and behavioural patterns for these two vectors, they share key common features: their distribution and abundance is intricately linked to climate-sensitive exposures and environmental conditions and can be affected by human activity such as urbanisation or deforestation, and they have both shown remarkable adaptability through behavioural plasticity, hybridization with other species, niche shifts, and other strategies [32–35]. However, the associations between climatic factors, mosquito abundance, and disease transmission are extremely complex and also depend on other factors such as vectorial capacity of the mosquito populations, human susceptibility to infection, population density at each location, and robustness of health systems [36,37]. One study conducted in Kenya assessing spatiotemporal overlapping of dengue, chikungunya and malaria found that co-occurrence was significantly associated with the presence of litter and crowded households [38]. Further, both *An. darlingi* and *Ae. aegypti* may be susceptible to similar control strategies, including use of insecticide-treated materials and indoor or outdoor insecticide spraying [39–46]. Thus, it is important to identify where and when these diseases can co-occur to better target integrated vector control programs and allocate resources more efficiently [47]. In addition, an integral assessment of vector control methods that are effective for both diseases is required, as patterns of insecticide resistance or susceptibility to other specific measures may be different for each species and in different areas [47].

Some limitations to our study should be acknowledged. First, we analysed surveillance data for laboratory-confirmed dengue and malaria, which may not reflect the total number of cases, as there are likely many clinically diagnosed cases without laboratory confirmation; additionally, there could be delays in reporting or bias in case ascertainment, especially among asymptomatic cases, which has been described especially for dengue. Second, as we mostly aimed at proposing a framework to describe spatiotemporal patterns of association, we did not consider other factors that may modify associations between dengue and malaria, such as human mobility, living conditions, weather, or sociodemographic factors. Despite these limitations, this study provides important insight into the complex associations between malaria and dengue transmission in a region in the Peruvian Amazon. Future studies assessing co-occurrence of these VBD in districts in which a strong association was observed may be warranted to elucidate other factors that may affect these associations. Understanding the common drivers of malaria and dengue and associations between these two diseases will be a fundamental part of mounting preventive and early response strategies that optimise resource allocation, especially as these diseases co-occur in increasingly more places throughout the tropics with climate change.

## Conclusions

Our findings show that there has been a marked increase in co-occurrence of dengue and malaria in Loreto over the last 22 years, with heterogeneous patterns of temporal association between the two diseases. We identified several districts in which there is a positive association between malaria and dengue cases and which could potentially benefit from integrated control programs for both diseases. Furthermore, the identification of four distinct patterns of associations between dengue and malaria cases in the region of Loreto, with some geographic clustering, can inform future studies focused on these specific areas that can help elucidate drivers and modifiers of these associations. Understanding the drivers of dengue and malaria is a crucial step in developing strategies to reduce morbidity and mortality from both diseases, especially as anthropogenic climate and environmental changes are increasing their incidence.

## Supporting information

**S1 Fig. Malaria and dengue case series by district in the Loreto department between 2000 and 2021.** (A) Crude weekly cases. (B) First difference of weekly cases.
(PNG)

**S2 Fig. Dengue incidence per 1000 people by district in the Loreto department between 2000 and 2021.** Maps produced in R v.4.5 using public data from Instituto Nacional de Estadística e Informática (INEI - Peru) contributors (https://estadist.inei.gob.pe/map) under Open Data Commons Open Database License (ODbL) 1.0 (http://openstreetmap.org/copyright).
(PNG)

**S3 Fig. Malaria incidence per 1000 people by district in the Loreto department between 2000 and 2021.** Maps produced in R v.4.5 using public data from Instituto Nacional de Estadística e Informática (INEI - Peru) contributors (https://estadist.inei.gob.pe/map) under Open Data Commons Open Database License (ODbL) 1.0 (http://openstreetmap.org/copyright).
(PNG)

**S4 Fig. Malaria and dengue incidence per 1000 people by district in the Loreto department between 2000 and 2021.**
(PNG)

**S5 Fig. Yearly maps for dengue and malaria co-distribution by district in Loreto between 2000 and 2021.** Dengue and malaria log-incidence (categorised using Fisher-Jenks algorithm for the entire study period). Maps produced in R v.4.5 using public data from Instituto Nacional de Estadística e Informática (INEI - Peru) contributors (https://estadist.inei.gob.pe/map) under Open Data Commons Open Database License (ODbL) 1.0 (http://openstreetmap.org/copyright).
(PNG)

**S6 Fig. TLCC analysis of dengue and malaria time series at lags ±2 years for all districts in Loreto.** ρ indicates the maximum correlation coefficient and lags are in weeks.
(PNG)

**S7 Fig. District-level TLCC results from cross-correlation analysis.** (A) TLCC coefficient by district (highest absolute value plotted). (B) Lag time in weeks for maximum TLCC coefficient by district. Maps produced in R v.4.5 using public data from Instituto Nacional de Estadística e Informática (INEI - Peru) contributors (https://estadist.inei.gob.pe/map) under Open Data Commons Open Database License (ODbL) 1.0 (http://openstreetmap.org/copyright).
(PNG)

**S8 Fig. Sensitivity analysis of time windows for TLCC.** TLCC analysis was repeated using time windows of 1–5 years. The maximum TLCC coefficient (scaled times 1000) and corresponding lag for each time window and district are plotted. Districts are categorised qualitatively into those with consistently positive coefficients (A), those with consistently negative coefficients (B), and those with inconsistent TLCCs or lags (C).
(PNG)

**S9 Fig. Map of qualitative categories from results of sensitivity analysis.** Maps produced in R v.4.5 using public data from Instituto Nacional de Estadística e Informática (INEI - Peru) contributors (https://estadist.inei.gob.pe/map) under Open Data Commons Open Database License (ODbL) 1.0 (http://openstreetmap.org/copyright).
(PNG)

**S10 Fig. Bivariate tile plot of sensitivity analysis.** Districts are ordered from highest to lowest TLCC coefficient in any of the five tested windows. Purple tiles had a positive TLCC coefficient and positive corresponding lag. Pink tiles had a positive TLCC coefficient and negative corresponding lag. Light blue tiles had a negative TLCC coefficient and positive corresponding lag. Light grey tiles had a negative TLCC coefficient and a negative corresponding lag.
(PNG)

## Author contributions

**Conceptualization:** Gabriel Carrasco-Escobar, Paloma M. Carcamo, Samantha R. Kaplan, Gordon C. McCord, Tarik Benmarhnia.

**Data curation:** Paloma M. Carcamo, Jesus M. Quispe.

**Formal analysis:** Paloma M. Carcamo, Jesus M. Quispe.

**Methodology:** Gabriel Carrasco-Escobar, Paloma M. Carcamo, Gordon C. McCord, Tarik Benmarhnia.

**Supervision:** Gabriel Carrasco-Escobar, Gordon C. McCord, Tarik Benmarhnia.

**Validation:** Jesus M. Quispe.

**Visualization:** Paloma M. Carcamo.

**Writing – original draft:** Paloma M. Carcamo.

**Writing – review & editing:** Gabriel Carrasco-Escobar, Paloma M. Carcamo, Samantha R. Kaplan, Jesus M. Quispe, Gordon C. McCord, Tarik Benmarhnia.

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
