## [Decision Letter · Decision Letter 0]

13 May 2025

PGPH-D-25-00114

Spatiotemporal co-distribution and time lagged cross correlation of malaria and dengue in Loreto, Peru

Dear Dr. Carcamo,

Thank you for submitting your manuscript to PLOS Global Public Health. After careful consideration, we feel that it has merit but does not fully meet PLOS Global Public Health’s publication criteria as it currently stands. Therefore, we invite you to submit a revised version of the manuscript that addresses the points raised during the review process.

The work is interesting, but it would be important to make the methods used more understandable by giving more suitable details, as suggested by reviewer 2. There is also the question of data origin and availability (site Regional Health Department of Loreto?)

We look forward to receiving your revised manuscript.

Kind regards,

Bernard Cazelles, Ph.D.

Academic Editor

Journal Requirements:

1. Please provide additional details regarding participant consent. In the ethics statement in the Methods and online submission information, please ensure that you have specified (1) whether consent was informed and (2) what type you obtained (for instance, written or verbal, and if verbal, how it was documented and witnessed). If your study included minors, state whether you obtained consent from parents or guardians. If the need for consent was waived by the ethics committee, please include this information.

2. Figure 1 and 3: please (a) provide a direct link to the base layer of the map (i.e., the country or region border shape) and ensure this is also included in the figure legend; and (b) provide a link to the terms of use / license information for the base layer image or shapefile. We cannot publish proprietary or copyrighted maps (e.g. Google Maps, Mapquest) and the terms of use for your map base layer must be compatible with our CC-BY 4.0 license.

Additional Editor Comments (if provided):

The work is interesting, but it would be important to make the methods used more understandable by giving more suitable details, as suggested by reviewer 2. There is also the question of data origin and availability (site Regional Health Department of Loreto?)

Reviewers' comments:

Reviewer's Responses to Questions

**Comments to the Author**

1. Does this manuscript meet PLOS Global Public Health’s publication criteria?

Reviewer #1: Partly

Reviewer #2: Yes

2. Has the statistical analysis been performed appropriately and rigorously?

Reviewer #1: Yes

Reviewer #2: No

3. Have the authors made all data underlying the findings in their manuscript fully available (please refer to the Data Availability Statement at the start of the manuscript PDF file)?

Reviewer #1: Yes

Reviewer #2: Yes

4. Is the manuscript presented in an intelligible fashion and written in standard English?

Reviewer #1: Yes

Reviewer #2: Yes

Reviewer #1: The manuscript was done correctly and the data supports the results presented and the highly relevant public health findings, however, I have a question:

Is the data used in the analysis from a public database? I couldn't find their source anywhere in the manuscript.

I also have some suggestions:

- I would suggest adding "Data sources" before "Study area" to provide greater clarity as to where the data were obtained.

- Include the R packages used to construct the maps, as they may be subject to copyright.

- In Figure 2, the titles should be better organized, as the position of "Lag (weeks)" is confusing to the reader.

- In sentences 274 to 277, the authors could better explain the environmental and geographic characteristics of the area and their associations.

Reviewer #2: The authors present a descriptive time series analysis of malaria and dengue incidence in Peru, with a particular focus on the Loreto province. Using cross-correlation and statistical tests, they investigate associations between the two time series across a range of temporal lags, at the district level. They identify several districts in Loreto where significant associations emerge between dengue and malaria incidence. The analysis is systematic, thoughtfully structured, and well presented. However, I have some concerns, especially regarding how the authors address seasonality in the time series, as this could introduce spurious correlations and bias the interpretation of temporal relationships. Below are several suggestions to further strengthen the manuscript:

Clarify the study rationale and hypothesis:

- The manuscript would benefit from a clearer statement of the research question and motivation. Why is it important to investigate time-lagged correlations between these diseases? What hypotheses are being tested? Clarifying the added value of this approach would help frame the relevance of the work.

Abstract clarity and specificity:

- The Abstract should explicitly state that the study compares dengue and malaria incidence.

- The sentence “The number of districts reporting both diseases has increased” appears abruptly. Consider a more informative formulation, e.g., “Our analysis shows a growing number of districts reporting both dengue and malaria cases over time.”

- Some phrases are overly technical for the Abstract, such as “We categorised districts based on dengue/malaria spatio-temporal patterns and conducted Moran’s tests...” A more accessible formulation could be: “We grouped districts based on shared temporal and spatial patterns in dengue and malaria incidence and explored how the two diseases co-occurred geographically.”

Introduction clarity:

- Line 66: “While these two VBD share many similarities” appears disconnected. The diseases should be clearly introduced earlier to give context.

- Line 99: The authors could be more precise and state that the study examines the incidence of the two diseases.

Methods transparency and consistency:

- The Methods mention weekly incidence, but Figure 1 presents monthly case counts. Please clarify how the monthly incidence was calculated.

- Figure 1 should be moved to the Results section, as it presents outputs from the descriptive analysis rather than methodological diagrams.

- Consider adding disaggregated incidence plots by district. These could reveal meaningful trends and enhance the interpretation of the district-level analyses.

Stationarity and preprocessing:

- Line 152 states that first differences were applied to ensure stationarity. It would be important to demonstrate that this step is sufficient (e.g., via the Augmented Dickey-Fuller test). Consider including supplementary plots of the differenced series for transparency.

More detail on analytical procedures:

- Please expand the Methods to include information on the significance thresholds used, the criteria for determining optimal lags, and how the Fisher-Jenks algorithm was applied.

- Critically, the manuscript should describe how seasonal patterns were handled. Seasonal differencing is essential when performing time-lagged cross-correlation on seasonal time series, to avoid spurious associations driven by periodic fluctuations.

Results interpretation:

- Clarify how districts in Figure 2 were selected—was this based on significance, magnitude of correlation, or another criterion?

- The conclusion that this framework could inform integrated control strategies is intriguing, but needs to be substantiated. The authors should specify what type of strategies could be informed and how, based on the current analysis. Are there operational implications for timing interventions or resource allocation?

Code availability:

- The GitHub repository is a valuable resource. It would be helpful to provide a short summary or guide in the README file outlining the structure of the code and how users can reproduce key analyses.

**Do you want your identity to be public for this peer review?** For information about this choice, including consent withdrawal, please see our Privacy Policy

Reviewer #1: No

Reviewer #2: No

---

## [Decision Letter · Decision Letter 1]

19 Nov 2025

Spatiotemporal co-distribution and time lagged cross correlation of malaria and dengue in Loreto, Peru

PGPH-D-25-00114R1

Dear Dr. Carcamo,

We are pleased to inform you that your manuscript 'Spatiotemporal co-distribution and time lagged cross correlation of malaria and dengue in Loreto, Peru' has been provisionally accepted for publication in PLOS Global Public Health.

Best regards,

Bernard Cazelles, Ph.D.

Academic Editor

Significant improvements have been done and now the manuscript can be published.

Reviewer Comments (if any, and for reference):

Reviewer's Responses to Questions

**Comments to the Author**

Reviewer #1: All comments have been addressed

Reviewer #2: All comments have been addressed

publication criteria?

Reviewer #1: Yes

Reviewer #2: Yes

3. Has the statistical analysis been performed appropriately and rigorously?

Reviewer #1: Yes

Reviewer #2: Yes

4. Have the authors made all data underlying the findings in their manuscript fully available (please refer to the Data Availability Statement at the start of the manuscript PDF file)?

Reviewer #1: Yes

Reviewer #2: Yes

5. Is the manuscript presented in an intelligible fashion and written in standard English?

Reviewer #1: Yes

Reviewer #2: Yes

Reviewer #1: I congratulate the authors on the final version of the article; I think it's a significant improvement and deserves to be published. The use of public data is now clear, as are many points in the article.

Reviewer #2: All my comments have been addressed, the manuscript looks more rigorous and provides all the necessary details as well as an adequate discussion.

**Do you want your identity to be public for this peer review?** For information about this choice, including consent withdrawal, please see our Privacy Policy

Reviewer #1: No

Reviewer #2: No
